# Roadblocks for Temporarily Disabling Shortcuts and Learning New Knowledge

**Hongjing Niu    Hanting Li    Feng Zhao*    Bin Li**
University of Science and Technology of China, Hefei, China
{sasori, ab828658}@mail.ustc.edu.cn, {fzhao956, binli}@ustc.edu.cn

## Abstract

Deep learning models have been found with a tendency of relying on shortcuts, i.e., decision rules that perform well on standard benchmarks but fail when transferred to more challenging testing conditions. Such reliance may hinder deep learning models from learning other task-related features and seriously affect their performance and robustness. Although recent studies have shown some characteristics of shortcuts, there are few investigations on how to help the deep learning models to solve shortcut problems. This paper proposes a framework to address this issue by setting up roadblocks on shortcuts. Specifically, roadblocks are placed when the model is urged to learn to complete a gently modified task to ensure that the learned knowledge, including shortcuts, is insufficient the complete the task. Therefore, the model trained on the modified task will no longer over-rely on shortcuts. Extensive experiments demonstrate that the proposed framework significantly improves the training of networks on both synthetic and real-world datasets in terms of both classification accuracy and feature diversity. Moreover, the visualization results show that the mechanism behind the proposed our method is consistent with our expectations. In summary, our approach can effectively disable the shortcuts and thus learn more robust features.

## 1  Introduction

Deep learning has triggered the current rise of artificial intelligence and is the workhorse of today's machine intelligence [13]. One of the great advantages of deep learning is its ability to automatically extract the required features according to the task. Moreover, the massive amount of data that drive the model is the key to its success. Regrettably, deep learning methods can hardly take full advantage of the training data. An important reason is its over-reliance on simple features. Recently, deep learning approaches have been found to have a tendency to learn simple features only [4]. The learned simple features may be sufficient for the accessible data (training data), and the model will have difficulty in learning other complex but effective features.

Insufficient learning of effective features may lower the accuracy and generalization performance. More severely, when the sample distribution changes and the learned simple features are no longer valid, deep learning models may collapse since they do not learn other features. Therefore, suppressing the reliance on simple features is indispensable for designing models.

Shortcuts in deep learning are described as the decision rules that perform well on standard benchmarks but fail to be transferred to more challenging testing scenarios according to [4]. We make a more specific explanation for easier evaluation. For a task, there are a number of features that should be considered. Since the training set is limited, perhaps only a subset of features is sufficient to make a decision. Decision rules that rely on an easy-to-learn subset of features are called shortcuts. The

---

*Corresponding author.

36th Conference on Neural Information Processing Systems (NeurIPS 2022).

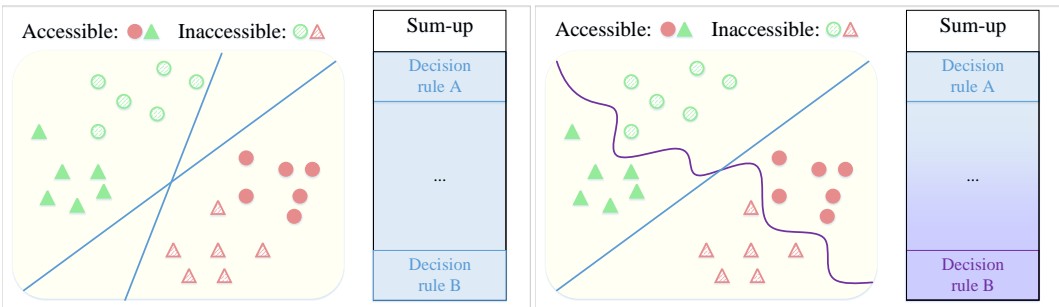

(a) Functionally similar decision rules.    (b) Functionally diverse decision rules.

Figure 1: Shortcuts and exhaustive decision rules.

concept of *shortcuts* has two key points: (1) only use a subset of effective features but perform well on the training set; (2) easy to learn and affect the learning of other complex but effective features.

Figure 1 provides an intuitive illustration of what shortcuts are and why shortcuts affect the generalization of deep learning models. Each sample in Figure 1 has two features (color, shape). The training set only has red circles and green triangles, while the testing set contains the other two combinations. For the training set, decision rules are valid whether they are based on color or shape, but both of them may fail on the testing set because the training set does not articulate the intention. A task may require the selection of a particular decision rule or a sum-up. But either way, the diversity of decision rules is a foundation.

Some studies expand the difference between decision rules at the feature level [10]. However, differentiated decision rules may still be functionally similar (see Figure 1a). To obtain the functionally diverse decision rules shown in Figure 1b, we propose to temporarily disable the shortcuts.

It is important to note that shortcuts are disabled temporarily, not forever. We believe that simplicity does not mean bad, but more decision rules might help. Our approach aims to simply provide more options. The integration of decision rules should depend on the task scenarios, and we also provide some suggestions.

To address the issue that deep learning methods do not learn anything other than shortcuts, we propose roadblocks to temporarily disable the shortcuts and force the model to learn new knowledge. The ability of roadblocks that disables the shortcuts is achieved by reconstructing datasets under the guidance of trained models. A reconstructed sample should retain most of the attributes of the original sample while disabling the learned attributes. Thus, the model trained with reconstructed samples will naturally reduce the reliance on shortcuts and learn new knowledge.

Enlarging the distance directly [10] at the feature level often only learns the decision rules shown in Figure 1a, but it is difficult to learn the functional differences. We believe the reason is the redundancy and weak interpretability of deep learning models. There is currently no metric that can measure the functional differences of the models well. Most previous researches on shortcuts focus on what features are more likely to be shortcuts, and the corresponding solutions mostly rely on the prior knowledge of shortcuts [5, 8, 18]. Priori-based methods require accurate prior knowledge and targeted design, which are difficult to accomplish in most scenarios. Unlike them, our roadblocks do not use any prior knowledge about shortcuts and can deal with different forms of shortcuts.

In summary, the contributions of this work are mainly three-fold:

- We propose a new framework to help models learn more diverse knowledge. Compared with those methods based on distance at the feature level, our approach can learn brand-new knowledge and obtain functionally diverse decision rules.

- We design a new strategy for disabling shortcuts, which requires no priors and is suitable for a wider range of application scenarios.

- Our method can effectively improve the generalization ability of the model and can reach state-of-the-arts in the debiasing task.

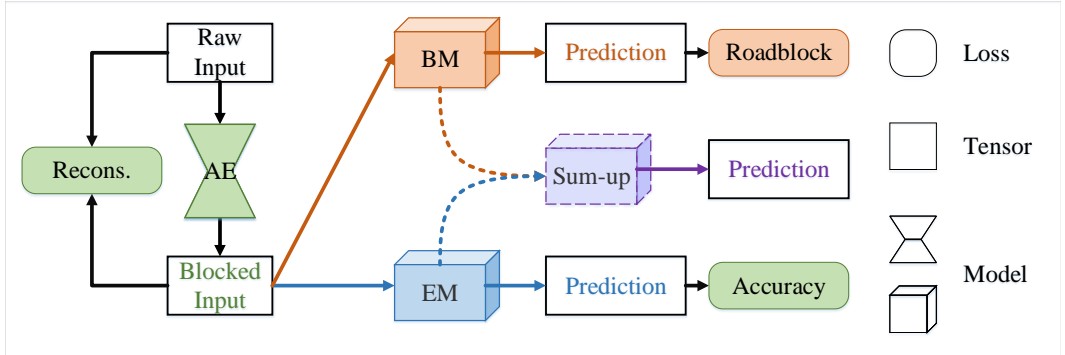

Figure 2: Framework of roadblocks. The basic framework of roadblocks consists of three sub-models: autoencoder (AE) in green, blocked model (BM) in red, and explorer model (EM) in blue. The purple model represents the sum-up of BM and EM.

## 2 Method

This section will introduce our strategy to address the shortcuts problem, which is temporarily disabling the shortcuts and learning more diverse knowledge. First, we specify some definitions and the evaluation system. Then, we introduce our roadblocks framework. Finally, the working process of the framework will be detailed.

### 2.1 Shortcut Suppression Task

Shortcuts are described as decision rules that perform well on independently-identical-distribution (i.i.d.) testing sets but fail to generalize to out-of-distribution (o.o.d.) testing sets [4]. We refer to this idea and describe the shortcut more precisely, **shortcuts are decision rules that rely on an easy-to-learn subset of features.**

For classification tasks, suppose a distribution $\mathcal{D}$ whose feature space can be represented by $\mathcal{F}$. Supposing that a training set $D_{train}$ is sampled from $\mathcal{D}$, and there exist a decision rule that based on $\mathcal{F}_{train}$ that can perform well on $D_{train}$. $\mathcal{F}_{train}$ is a subspace of $\mathcal{F}$. Such a decision rule has the potential risk of failing in other datasets sampled from $\mathcal{D}$ because it ignores other features in $\mathcal{F}$.

For ease of measurement, we refer to the setting of the debiasing task. The feature space contains two main features, $\mathcal{F} = [F_T, F_B]$. In the training data, the target feature $F_T$ corresponds to the ground truth perfectly; while biased feature $F_B$ corresponds to the ground truth with a large probability. Nonetheless, there is still a possibility that $\mathcal{F}_{train}$ is closer to $F_B$ because $F_B$ is simpler, which means that the model may not have learned $F_T$ as we expected. To verify the learning of $F_T$ in the testing data, $F_T$ still corresponds to the ground truth perfectly, and $F_B$ is no longer helpful.

### 2.2 Framework of Roadblocks

Assuming that existing models use shortcuts, a strategy to find more reasonable decision rules is to use the idea of roadblocks that forces the new model not to use the same decision rule as any previous model.

Figure 2 illustrates the framework of our approach. The overall model is composed of three parts. Blocked models are trained models that perform well on the i.i.d. test dataset. The autoencoder is used to modify the original input, thereby limiting the performance of blocked models. Once the blocked models perform poorly on the modified input, the explorer models trained with the modified input will find new ways to achieve competitive performance. None of the three sub-models need a specific structure, as long as they have sufficient representation capabilities.

Let AE, BM, EM denote autoencoder, blocked model, explorer model, respectively. BM is a pre-trained model with good performance, which is fixed during the training process. AE receives the raw input $I_{raw}$ and modifies it into blocked input $I_{rec}$. The mean square error (MSE) is used to measure

the similarity between them,

$$Loss_{rec} = MSE(I_{raw}, I_{rec}).$$ (1)

The reconstructed input needs to meet two requirements, (1) it is difficult to distinguish with the existing decision rules, and (2) it can be correctly distinguished.

$I_{rec}$ will first be used as the input of BM. The MSE of the prediction $BM(I_{rec})$ and the pseudo label $label_{ep}$ will be used as the roadblock loss,

$$Loss_{rb} = MSE(BM(I_{rec}), label_{ep}).$$ (2)

The pseudo label $label_{ep}$ is set to consider all possible categories with equal probability, which is $label_{ep} = [\frac{1}{n}, \frac{1}{n}, ..., \frac{1}{n}])$, where $n$ is the number of categories. In this way, the purpose of AE is to generate $I_{rec}$ to confuse BM.

$I_{rec}$ will also be used as the input of EM. In order for the EM to achieve better classification performance, the accuracy loss between the prediction $EM(I_{rec})$ and the ground-truth label is calculated for training,

$$Loss_{ac} = MSE(EM(I_{rec}), label_{gt}).$$ (3)

## 2.3 Working Process

**Core training process** BM is a pre-trained model, and it does not participate in the training process of the proposed framework. Both AE and EM need to be trained. We alternately train them for a more stable training process. A typical training cycle consists of two parts, (1) AE is trained with roadblock loss and reconstruction loss; (2) EM is trained under the guidance of accuracy loss,

$$\theta_{AE} = \underset{\theta_{AE}}{\arg\min}(\lambda_{rb}Loss_{rb} + \lambda_{rec}Loss_{rec}),$$ (4)

$$\theta_{EM} = \underset{\theta_{EM}}{\arg\min}(\lambda_{ac}Loss_{ac}).$$ (5)

Here, $\theta_{AE}$ and $\theta_{EM}$ denote the parameters of AE and EM respectively, and $\boldsymbol{\lambda} = [\lambda_{rb}, \lambda_{rec}, \lambda_{ac}]$ are the corresponding loss weights.

**Multi-model** BM can be any model that performs well on training set. In addition to vanilla models, a EM generated by roadblocks method can also be used as BM to train a new EM. BM can also be a set of models $BM = [BM_1, BM_2, ..., BM_n]$. Correspondingly, the roadblock loss needs to be adjusted to

$$Loss_{rb} = \frac{1}{n} \sum_i MSE(BM_i(I_{rec}), label_{ep}).$$ (6)

We suggest adding the well-trained EM to the set of BM and then training the new EM. When the new EM fails to perform well on the training set, it means that it is difficult to learn more effective features, at which point the loop terminates. The experiments suggest that in most cases, two models are sufficient. Therefore, to reduce the computational cost, we usually use two models.

**Sum-up** The proposed framework contains multiple sub-models that learn different effective features respectively. We do not recommend discarding any features directly, as the ease of learning and the effectiveness of features are not necessarily related. We recommend designing according to specific scenarios. Without priors on the test scene, we recommend using the feature fusion strategy because it is more likely to have good generalization. Another idea is to vote, and o.o.d. samples can be detected if there is a significant difference in the votes of the submodels. For some tasks with priors, the appropriate model can be directly selected. For example, for debiasing tasks, a reasonable idea is to choose the model that performs better on the training set because the biased features are less matched with the ground truth.

## 3 Experiments

To verify whether functionally different decision rules are learned, we choose datasets with multiple attribute labels for experiments. The experimental setup refers to the design of the debiasing task [15]. Each individual attribute in the training set corresponds well to the ground truth, while only one attribute matches the ground truth one-to-one in the testing set. Taking Figure 1 as an example,

Table 1: Image classification accuracy evaluated on CMNIST datasets. The ratio represents the proportion of minority samples, and the smaller the ratio is, the more difficult it is to debias. The best results are highlighted in bold, while the second-best results are denoted with underlines. A check mark (✓) indicates that no priors about the feature are used, while a cross mark (×) means the opposite. The results of previous methods are from [15].

| Dataset | Ratio (%) | Vanilla [7] ✓ | EnD [29] × | ReBias [1] × | LFF [23] ✓ | LDD [15] ✓ | Ours ✓ |
|---|---|---|---|---|---|---|---|
| CMNIST | 0.5 | $35.19_{\pm3.49}$ | $34.28_{\pm1.20}$ | $\mathbf{70.47}_{\pm1.84}$ | $52.50_{\pm2.43}$ | $65.22_{\pm4.41}$ | $\underline{66.64}_{\pm2.15}$ |
| | 1.0 | $52.09_{\pm2.88}$ | $49.50_{\pm2.51}$ | $\mathbf{87.40}_{\pm0.78}$ | $61.89_{\pm4.97}$ | $81.73_{\pm2.34}$ | $\underline{82.04}_{\pm1.61}$ |
| | 2.0 | $65.86_{\pm3.59}$ | $68.45_{\pm2.16}$ | $\mathbf{92.91}_{\pm0.15}$ | $71.03_{\pm2.44}$ | $84.79_{\pm0.95}$ | $\underline{84.93}_{\pm1.34}$ |
| | 5.0 | $82.17_{\pm0.74}$ | $81.15_{\pm1.43}$ | $\mathbf{96.96}_{\pm0.04}$ | $80.57_{\pm3.84}$ | $\underline{89.66}_{\pm1.09}$ | $88.65_{\pm0.93}$ |

Table 2: Debiasing performance on CelebA. "Unbiased" means that all combinations appear at the same frequency, and "Bias-conflicting" indicates that only the minority of combinations appear. The † results are from [15] and ‡ results are from [29].

| Dataset | Testing set | Vanilla [7] ✓ | EnD [29] × | ReBias [1] × | LFF [23] ✓ | LDD [15] ✓ | Ours ✓ |
|---|---|---|---|---|---|---|---|
| CelebA | Unbiased | $62.00^{\ddagger}_{\pm0.02}$ | $\underline{75.93}^{\ddagger}_{\pm1.31}$ | $69.33_{\pm1.72}$ | $66.20^{\ddagger}_{\pm1.21}$ | $64.82_{\pm1.36}$ | $\mathbf{79.73}_{\pm1.15}$ |
| | Bias-conflicting | $33.75^{\ddagger}_{\pm0.28}$ | $\underline{53.70}^{\ddagger}_{\pm5.24}$ | $44.20_{\pm7.09}$ | $45.48^{\ddagger}_{\pm4.33}$ | $47.63_{\pm4.58}$ | $\mathbf{64.06}_{\pm0.78}$ |
| BAR | Bias-conflicting | $49.91^{\dagger}_{\pm0.53}$ | $52.16^{\dagger}_{\pm0.18}$ | $53.25^{\dagger}_{\pm0.63}$ | $58.81^{\dagger}_{\pm2.64}$ | $\underline{63.50}^{\dagger}_{\pm1.47}$ | $\mathbf{69.51}_{\pm2.43}$ |

the training set is mainly red circles and green triangles, and the four combinations have the same proportion in the testing set. We use the synthetic dataset CMNIST [15], which adds a second attribute by coloring MNIST [14]. We also use real-world datasets CelebA [20], and BAR [23] that were validated to have shortcuts to test the practicality of our method. Details of datasets and implementation are described in the appendix. Extensive experiments have been conducted to demonstrate the following effects of roadblocks: 1) greatly enhance the ability to explore new knowledge, 2) significantly improve the generalization of deep learning models.

## 3.1 Exploration of Knowledge

To verify whether roadblocks can reduce models' reliance on shortcuts, we compared the performance of the conventionally trained CNNs (Vanilla) and some representative debiasing methods and our roadblocks method. We first test on a controlled datasets CMNIST to facilitate analysis. The experimental results are shown in Table 1.

For CMNIST, each sample has two key features, color, and digit. Although the colors are slightly less matched with ground truth, deep learning models tend to learn colors. Our method can help deep learning models learn more decision rules that utilize digit features.

Models also often rely on shortcuts to complete their tasks in real-world scenarios. We conducted experiments on the face attribute dataset CelebA [20] and the action recognition dataset BAR [23]. For real-world datasets, we keep its native ratio to maintain the authenticity during the training process. And then we test whether the model learned more knowledge through testing sets of different distributions. For CelebA, following the experimental setup of LFF [23], we trained models to classify 'HeavyMakeup'. Then we select the same number of samples from all possible combinations of [Gender, HeavyMakeup] to form the unbiased testing set and minority combinations to form the bias-conflicting testing set. BAR contains six actions, and the backgrounds of the training and testing sets are significantly different. The testing set has only bias-conflict samples. The results are shown in Table 2. It can be seen that our method has obvious performance improvement compared to the state-of-the-art methods. In particular, for the bias-conflict dataset which is more difficult, our method achieves more than 10% improvement.

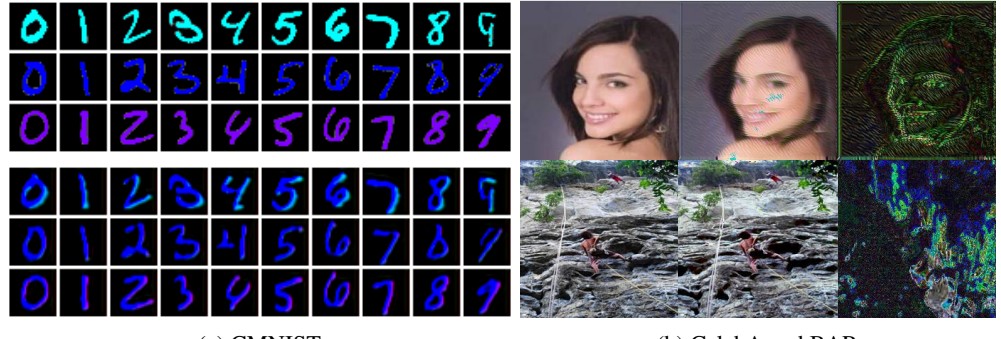

(a) CMNIST            (b) CelebA and BAR

Figure 3: Visualization of reconstructed images. For (a) CMNIST, the top 3 rows shows some of the original samples, and bottom 3 rows depict the modified samples generated during the training process. For (b) CelebA and BAR, the left, middle, and right are original images, reconstructed images, and their differences. Results are picked randomly, more results can be seen in the Appendix.
.

## 3.2 Disabling Shortcuts

The above experiments verified the effectiveness of our roadblocks framework in learning functionally diverse decision rules. Next, it is necessary to confirm whether the effective mechanism of the proposed method is consistent with the original intention of the design. Specifically, it is necessary to verify that some information in the reconstructed image is indeed corrupted so that the existing decision rules cannot make correct predictions. Thus, we visualized reconstructed images, and the result is shown in Figure 3. It can be observed that for CMNIST, our method can implement the disabling of the shortcut (color) by repainting digits. For different digits, the strategy of repainting is consistent, which shows that only the color attribute is disabled in the painting stage, while the number attribute is retained. Therefore, it is still feasible to use these samples to learn numbers, which is also well illustrated in Table 1.

For real-world datasets, similar modification methods are challenging to achieve, and our approach mainly implements the destruction by adding noise. For CelebA, AE tries to blur the gender attribute by some blurring (mainly focused on the face contour). For BAR, the blurring of AE mainly revolves around the environment. For the convenience of observation, we show the difference map of the original image and the reconstructed image, which is five times the absolute value of the difference of each pixel value between the two.

We also used t-SNE [30] to visualize the features extracted by the vanilla model and our approach, and the result is shown in Figure 4. Each colored number represents a sample. It is evident that the vanilla model does an excellent job of partitioning the dataset according to color. However, a closer look reveals that samples of the same color, regardless of the digit, are in the same cluster. In contrast, the model trained by our approach can distinguish digits and have certain invariance to color. Distinguishing digits is more complex than color, so getting sharp boundaries is more complicated.

To visually demonstrate the model's ability to explore new knowledge, we use Grad-CAM [26] to visualize the focus of models on the BAR dataset, and saliency maps are shown in Figure 5. In this experiment, we use ResNet-18 as the backbone, and the last fully connected layer is modified to a linear layer (512, 10) to match the number of categories. And the last convolutional layer is used as the target layer of Grad-CAM. For CMNIST, it is difficult to tell the difference from the saliency map because the colors and shapes correspond to the same regions. This is also the reason why we do not disable shortcuts using saliency map based methods. For CelebA and BAR, it can be seen that the attention regions of our approach are significantly different from that of the vanilla model. For correctly classified samples, our approach can find more effective information. Grad-CAM also provides some explanation for samples that vanilla misclassifies, but our method classifies correctly. For example, for samples of mountain climbing, the EM of our approach focuses on the climber, and the vanilla model focuses on the mountain. Both of these features are important, and our approach has better generalization ability because it utilizes more effective features. The samples of CMNIST and CelebA were randomly selected, and we selected more representative samples for BAR. There

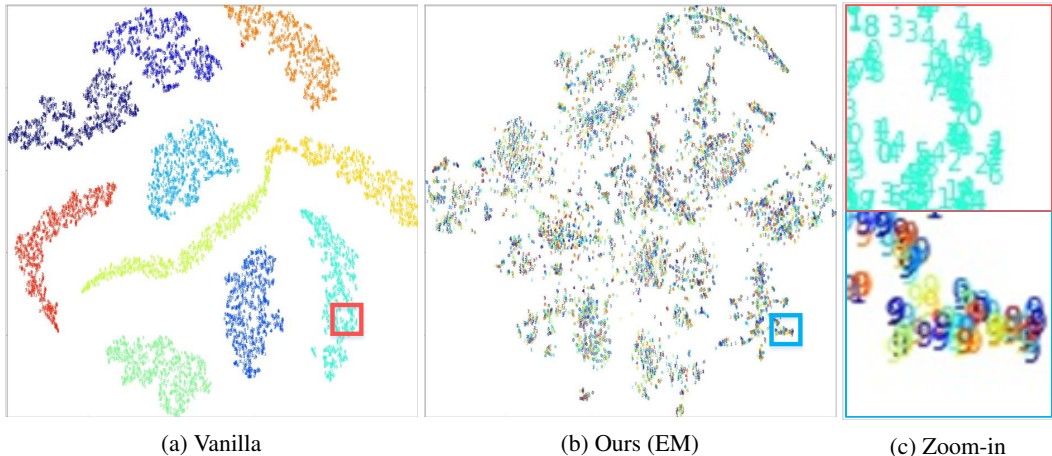

(a) Vanilla        (b) Ours (EM)        (c) Zoom-in

Figure 4: The t-SNE results on CMNIST. It reflects the distance between the features extracted by the model to some extent. Each colored number represents a sample. (a) corresponds to the vanilla model, (b) corresponds to our model, and (c) provides a zoom-in view of the boxed areas.

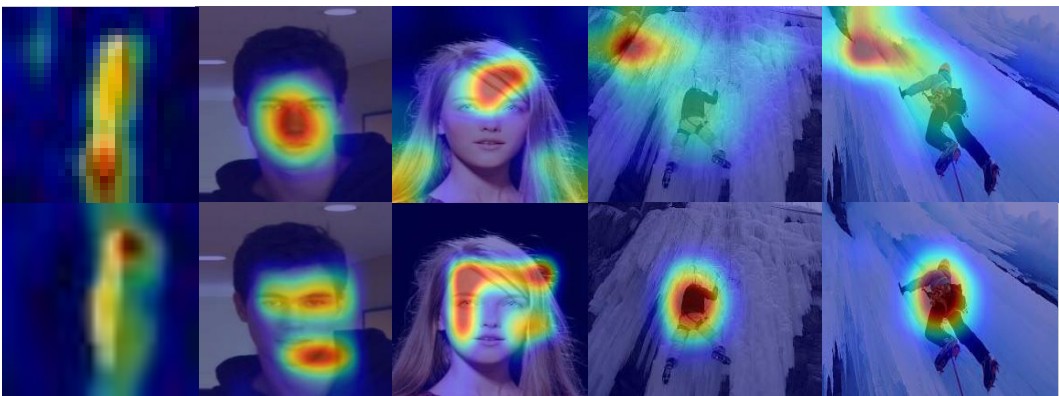

Figure 5: Saliency maps. The top and bottom rows show the Grad-CAM results of vanilla model and EM of our approach, respectively. Warm colors indicate a high level of attention. Green borders mean correct classification, while red borders imply incorrect classification.

are more examples of samples in the Appendix to facilitate more analysis and discussion, while also ensuring generality.

Grad-CAM is an excellent interpretability method, but it can only show differences in models at a spatial scale. Therefore, for datasets such as CMNIST, where different features overlap a lot in spatial scales, saliency maps cannot well reflect the differences between models.

The reconstruction results and saliency maps visually demonstrate that our approach does learn more effective features following our design intention. These methods still have some limitations. To verify the ability of our method to improve model diversity as fully as possible, we further quantify the model diversity.

### 3.3 Model Diversity

Our core purpose is to learn more knowledge (i.e., effective features). Therefore, it is necessary to choose appropriate metrics to measure the similarity between models. We applied SVCCA [24] to measure the similarity of the models. SVCCA is a tool for comparing two representations in a way that is invariant to affine transform. It receives the feature maps of middle layers and calculates the similarity between each pair of feature maps. Figure 6 shows some SVCCA results between the models. Figure 6a and Figure 6b are calculated on CMNIST (5pct), Figure 6c and Figure 6d

are calculated on BAR. CelebA is too large to be calculated, so we did not perform corresponding experiments.

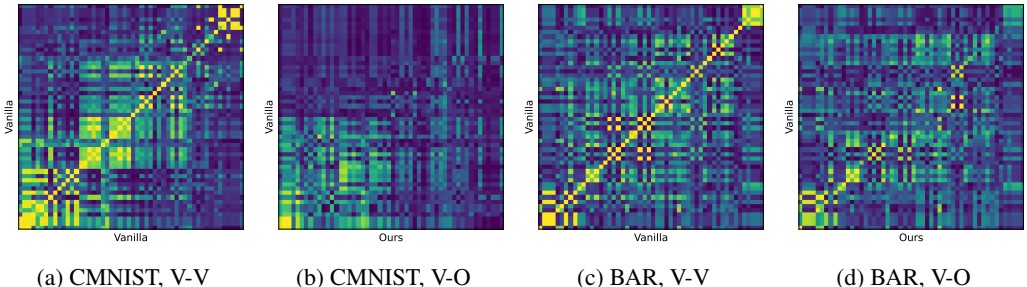

(a) CMNIST, V-V     (b) CMNIST, V-O     (c) BAR, V-V     (d) BAR, V-O

Figure 6: The SVCCA similarities. Bright tones indicate higher similarity. 'V' and 'O' denote vanilla and ours respectively. For example, (a) represents the similarities between layers of two vanilla models on the CMNIST dataset.

For the CMNIST dataset, the diagonal of the SVCCA matrix of the vanilla model is bright, indicating a high similarity between the same layers. While the similarity between our model and the vanilla model is slight, there are only a few brighter regions between several lower layers. The comparison of the two shows that the model trained by our method is significantly different from the vanilla model at the feature level. Therefore, our method helps to improve the diversity of models. It can also be observed that some adjacent layers also have high similarities. The BAR dataset is more complex than the CMNIST dataset. Therefore, the similarity between adjacent layers in the vanilla model is lower. Some layers of our model and the corresponding layers vanilla model are similar, which we think may correspond to some complex semantic features.

## 4 Related Work and Discussion

### 4.1 Related Work

**Shortcut learning**. Geirhos et al. [4] provided a comprehensive summary of shortcuts and defined them as decision rules that perform poorly on the o.o.d. testing set. A considerable amount of research has been devoted to analyzing the formation of shortcuts, providing many valuable conclusions [27, 22]. Deep learning models is found to have the tendency to use textures rather than shapes to make predictions [5], which has inspired a new round of thinking among researchers on the mechanism of deep learning. Katherine and Andrew [9] explored the conditions under which features are more likely to be shortcuts, which is very helpful to deepen our understanding of shortcuts. In addition, various data augmentation methods are used to drive models to learn features beyond known shortcuts. Some researches [5, 8, 18] assumed that the texture feature is the only shortcut, and the designed data augmentation techniques effectively reduce the model's reliance on texture. There are also ways to exclude shortcuts by introducing a prior on the difficulty of the target task [3]. However, research on suppressing shortcut without relying on prior knowledge is still basically an unknown area to explore.

**Fairness and debiasing**. The debiasing task can be understood as solving the problem that the model has learned a certain bias attribute in the data set, but failed to learn the relatively difficult target attribute. Some researchers have explored the preferences of deep learning methods themselves and dealt with them accordingly, such as the tendency to use textures rather than shapes [5] and the tendency to learn low-frequency features in images [2]. Besides, several existing approaches aimed at mitigating the bias also assume certain bias types [12, 31, 17, 29, 25]. Instead of defining specific types of bias, some approaches [23, 15, 19, 16] rely on the straightforward assumption that simple features are biased features. A typical example is *Learn from Failure*. The main idea is twofold; (a) intentionally train the first network to be biased by repeatedly amplifying its "prejudice", and (b) debias the training of the second network by focusing on samples that go against the prejudice of the biased network in (a). Unlike them, our approach does not use biased priors, but solves the debiasing problem by learning more knowledge. Compared with other debiasing methods, it has a wider range of application scenarios.

**Adversarial training** The proposed roadblocks share some similarities with adversarial training. Adversarial examples obtained by adding small designed perturbations can make the model go wrong [28]. Adversarial training [11] can effectively improve the adversarial robustness of the model. Our approach and adversarial training clearly differ in purpose, which also introduces design differences. Adversarial examples only require the model to predict incorrectly, whereas our method requires the model to be completely confused. Therefore, by design, the false labels we use are all categories, including the true category, with equal probability. We also conduct experiments using adversarial training, and the results are shown in the Appendix.

## 4.2 Discussion

**Functionally identical features may have different representations.** Increasing the diversity of features, in other words, learning more knowledge, is a natural idea, but its implementation is not as simple. An intuitive way to learn different features is to expand the distance between the two models at the feature level. We refer to the framework of negative correlation learning and expand the distance between models at the feature level as much as possible on the premise of ensuring consistent prediction results. Experimental results (see the Appendix) show that it is difficult to learn decision rules other than shortcuts through such a learning paradigm. We believe that the reason is that the excellent representational ability of the deep learning model allows it to represent functionally identical features in different ways. Therefore, a simpler and more effective approach is our roadblocks, which forces the model to learn new features by temporarily disabling the already learned features by modifying the input. Rather than some distance between features, we remind that functional differences between features may be more important.

**From comprehensiveness to exactness.** Our method is designed for general image classification tasks. The priors about the features and the distribution difference between training and testing sets are unknown. On the one hand, comprehensiveness (i.e., maximizing the use of effective features in the accessible data) is a reasonable goal since such models are the most robust against complex test scenarios. For comprehensiveness, our approach allows the model to temporarily disable shortcuts and learn new effective features as much as possible. Its effectiveness and benefits have been described above. On the other hand, exactness (i.e., only using some specific features) is a more realistic goal for many tasks. Our approach also provides a new implementation for feature screening. By first training a model to learn features that are not required for the task and then restricting the use of such features, it will be easier for the model to follow the designer's intention. Achieving the requirement for exactness at the lowest cost will be a key point for the designer.

## 5 Conclusion

In this work, we develop a novel framework to prevent deep learning models from merely using shortcuts (simple decision rules) while ignoring other effective features. Experimental results on multi-attribute datasets in complex testing scenarios demonstrate that our roadblocks have a strong capacity of reducing reliance on shortcuts and learning new knowledge, which is very difficult for conventional deep learning methods and debiasing models. Extensive experiments also show that roadblocks achieve excellent performance in terms of extracting new knowledge and improving model diversity. We expect our roadblocks can help shed light on the nature of shortcuts learning in neural networks, and provide a feasible paradigm for training models without depending on shortcuts.

## 6 Acknowledgments

This work was supported by the Anhui Provincial Natural Science Foundation under Grant 2108085UD12, and in part by the National Natural Science Foundation of China under Grants U19B2044 and 61836011. We acknowledge the support of GPU cluster built by MCC Lab of Information Science and Technology Institution, USTC.

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
