# A  Appendix

## A.1  Implementation Details

Our code is mainly based on PyTorch and will be available at `https://github.com/badcode-iip/Roadblocks`. The datasets used in the paper are all public datasets. There are many versions of CMNIST, the one we use is the version as LFF [23]. It contains identically distributed training and validation sets, and an unbiased testing set. We did not use any preprocessing for CMNIST. The details and download methods of CelebA [20] and BAR [23] can be found in the corresponding papers. We preprocessed CelebA and BAR (resize, normalize) for easier training, we also applied horizontal flipping for augmentation. The implementation of t-SNE is from scikit-learn, and the implementations of Grad-CAM and SVCCA are all from open source code. For Table 1, we use 3-layer MLPs, the number of neurons are (3*28*28, 100, 100, 32, 10). For the rest of the experiments, we use ResNet-18 as the classifier, and the pre-training parameters provided by PyTorch only in experiments on CelebA and BAR. The autoencoder is consist of a encoder (3 Conv2d layer) and a correspond decoder. We use Adam as the optimizer, learning rate is 1e-3 and alpha is 1e-6. The max epoch is 100. The batch size is 64 for CMNIST and 16 for CelebA and BAR. $[\lambda_{ac}, \lambda_{rb}, \lambda_{rec}]$ are set to [1, 1, 10] for CelebA and [1, 0.01, 10] for CMNIST and BAR.

## A.2  Functionally identical features may have different representations.

We use negative correlation learning to train models with feature-level differences, hoping to learn different decision rules. In order to show the results more intuitively, we divide the testing set into independently-identical-distribution (i.i.d.) part and out-of-distribution (o.o.d.) part. The results are shown in Table 3. It can be seen that none of them can classify o.o.d. samples correctly, which means that none of them can learn features other than color.

Table 3: Performance of different knowledge exploring methods.

| Method | Metric | i.i.d. set | o.o.d. set |
|--------|--------|------------|------------|
| NCL | L1 | $1.00 \pm 0.00$ | $0.00 \pm 0.00$ |
| NCL | L2 | $1.00 \pm 0.00$ | $0.00 \pm 0.00$ |
| NCL | JS | $1.00 \pm 0.00$ | $0.00 \pm 0.00$ |
| NCL | Wasserstein | $1.00 \pm 0.00$ | $0.00 \pm 0.00$ |

## A.3  What if autoencoder is not strong enough?

The reconstruction capability of AE has a direct impact on the setting of roadblocks. In this section we use 3-layer MLPs to build AE. By adjusting the reconstruction ability and the weight of reconstruction, we conduct further analysis on the behavior of the model. While keeping other hyperparameters unchanged, we modified the number of neurons in the code layer of AE to limit its reconstruction capability, then we repeated the previous experiment, the fluctuation of performance is shown in Figure 7a.

When the reconstruction capability of AE is low, the model cannot complete the basic i.i.d. task, and when it is high, the model may again prefer shortcuts. Only the results in the middle section just meet our expectations. Fortunately, we can avoid the first case by requiring reasonable training accuracy. And for the second case, we propose two simple strategies, on the one hand, we can consider the setting that just avoided the first case; on the other hand, adjusting the weights of reconstruction loss can also guide models with high reconstruction capabilities to make reasonable trade-offs.

Let us change a perspective and discuss the behavior of AE. It can be seen that with the expansion of the code layer, the accuracy on the i.i.d. testing set continues to improve, and the accuracy on the o.o.d. testing set increases first and then decreases. When the reconstruction capability is insufficient, AE tends to use attributes other than shortcuts, which is understandable because the roadblock loss is preventing AE from using shortcuts. As the reconstruction capability improves, AE gradually learns shortcuts and weighs the shortcuts and other attributes under the guidance of losses.

When the ability to rebuild is greatly compromised, the weight of reconstruction loss also has impact on the training process. We use hold the structure of BM and EM as ResNet-18, the weights of

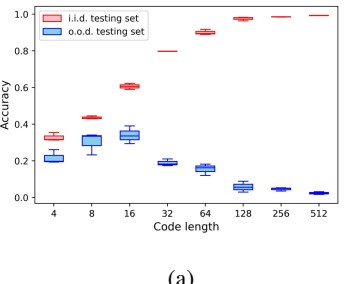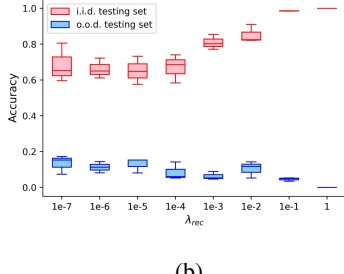

| (a) | (b) |

Figure 7: The impact of changes in code length and $\lambda_{rec}$.

other losses $\lambda_{rb}$ and $\lambda_{ac}$ are both set to 1. The structure of AE is MLP, (512, 256, 512). When other hyperparameters remain unchanged, the experimental results of different $\lambda_{rec}$ are shown in Figure 7b. The settings of $\lambda_{rec}$ can be roughly divided into three types according to i.i.d. set accuracy and o.o.d. set accuracy. (1) too high, $Acc._{i.i.d.}$ is close to 1, and $Acc._{o.o.d.}$ is close to 0, which means that AE tends to completely reconstruct original images, and EM can still learn shortcuts since roadblocks are invalid; (2) too low, $Acc._{i.i.d.}$ is low, and $Acc._{o.o.d.}$ is close to a random result, although roadblocks can work at this time, it is difficult for EM to learn with reconstructed images; (3) appropriately, $Acc._{i.i.d.}$ is good enough, and $Acc._{o.o.d.}$ is close to a random result, which indicates that EM has learned attributes other than shortcuts, and no longer relies too much on shortcuts while maintaining accuracy. It can be seen that when $\lambda_{rec}$ is not too high, $Acc._{o.o.d.}$ is stable, so even if we may not be able to obtain o.o.d. testing sets in real scenarios, it is not difficult to find suitable $\lambda_{rec}$ under the guidance of $Acc._{i.i.d.}$.

## A.4   Can adversarial training handle shortcuts?

We discuss the relationship of our method to adversarial training in the main text. Table 4 is the performance of simple adversarial training with fast gradient sign untargeted adversarial attack (FGSM) [6] under the l2 constraint on CMNIST. The maximum norm of the adversarial perturbation $\epsilon$ is 0.3, we use Adam as the optimizer, learning rate is 1e-2. It can be seen that adversarial training does not have a significant effect on biased datasets.

Table 4: Adversarial training on CMNIST.

| Ratio (%) | Vanilla | Adversarial |
|-----------|---------|-------------|
| 0.5 | $74.90 \pm 2.10$ | $72.78 \pm 4.48$ |
| 1 | $65.77 \pm 1.11$ | $64.14 \pm 1.41$ |
| 2 | $50.13 \pm 2.18$ | $45.31 \pm 4.23$ |
| 5 | $33.07 \pm 0.87$ | $30.47 \pm 0.90$ |

We further conducted experiments under different adversarial training parameter settings, and the results are shown in Table 5. It can be seen that the accuracy of the adversarial trained models on the adversarial samples and the clean samples are basically the same. This is a demonstration that adversarial training has helped the model achieve a certain degree of adversarial robustness. As epsilon increases, the accuracy of l2 improves, while the accuracy of linf increases first, but it is difficult to converge when epsilon is too large.

Some visualizations of adversarial examples shown in Figure 8 can help us understand why adversarial training has limited effect on the shortcut problem. It can be seen that the difference between the adversarial sample and the real sample is almost invisible to the naked eye, and it is difficult to observe clear digital information in the adversarial noise magnified by 100 times.

We also conduct experiments on project gradient descent (PGD) [21], but it seems difficult to obtain effective adversarial examples using MLPs. So we use ResNet-18 as the backbone. epsilon=0.03, k=4, alpha=0.03/4. The accuracy is 92.91 on CMNIST (5pct) and 88.54 on adversarial examples.

Table 5: The performance of adversarial training on CMNIST under different experimental settings.

| $\epsilon$ | 0.001 | 0.005 | 0.01 | 0.05 | 0.1 | 0.5 |
|---|---|---|---|---|---|---|
| l2-clean | 68.45 | 66.13 | 71.03 | 73.37 | 71.59 | 76.69 |
| l2-adv | 68.44 | 66.06 | 70.97 | 73.61 | 70.97 | 73.19 |
| linf-clean | 69.47 | 73.66 | 77.26 | 67.70 | 51.58 | 11.35 |
| linf-adv | 69.44 | 73.14 | 74.00 | 52.08 | 35.85 | 11.35 |

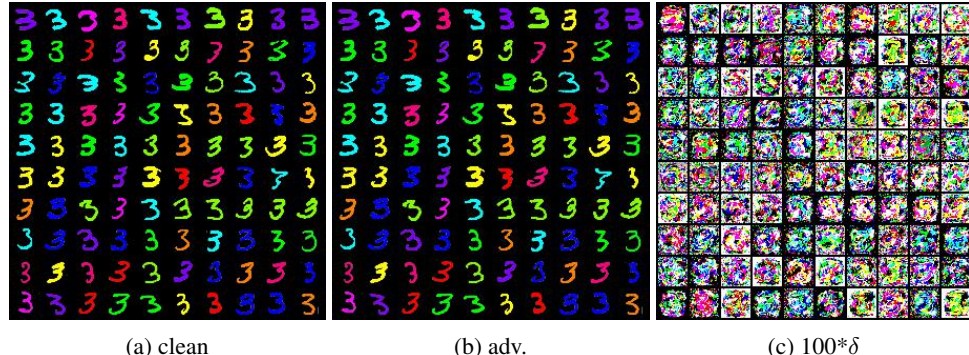

(a) clean         (b) adv.         (c) 100*$\delta$

Figure 8: FGSM samples. Results are taken from the testing set of CMNIST ($ratio = 5\%$).

(vanilla can achieve 93.34). Figure 9 shows some samples of PGD. It can be seen that the noise of PGD seems to be more uniform compared to that of FGSM. Neither can distinguish colors or numbers with the naked eye.

Based on these experimental results, we believe that simple adversarial training cannot solve the shortcut problem that this paper attempts to solve.

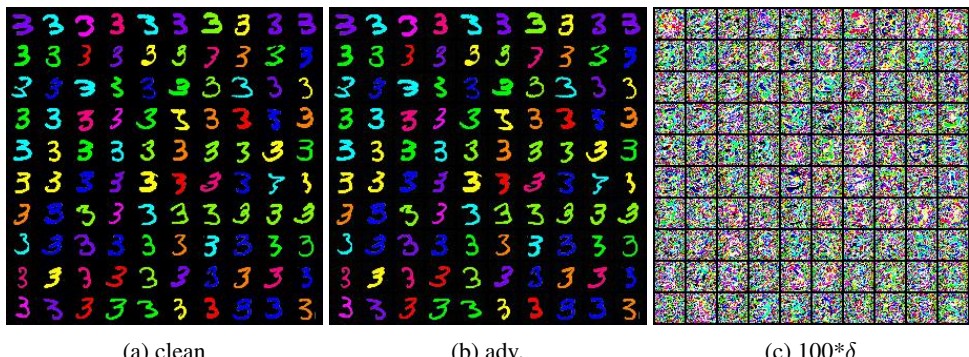

(a) clean         (b) adv.         (c) 100*$\delta$

Figure 9: PGD samples. Results are taken from the testing set of CMNIST ($ratio = 5\%$).

### A.5 How to choose hyperparameters?

The weight of loss function is a group of important hyperparameters. Based on empirical values, we adjust the loss function curve during the experiment to ensure that several objectives can be optimized. We tested different combinations of lambdas on the BAR dataset and selected the one that performed better, the result is shown in Figure 10. We refer to LDD's settings [15] for optimizer.

### A.6 More visualization results

We show more visualization experiments to demonstrate the generality of our conclusions. Figure 11 is the result of CelebA and Figure 12 is the result of BAR. For both of them, from top to bottom, is

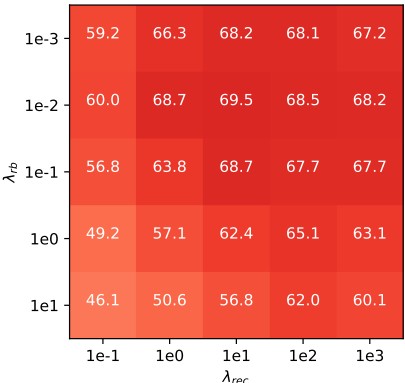

Figure 10: The sensitivity to weights of the losses. The numbers represent the testing accuracy, and a darker color represents a higher accuracy.

the original image, the reconstructed image, the difference, the saliency map of the original model, and the saliency map of EM.

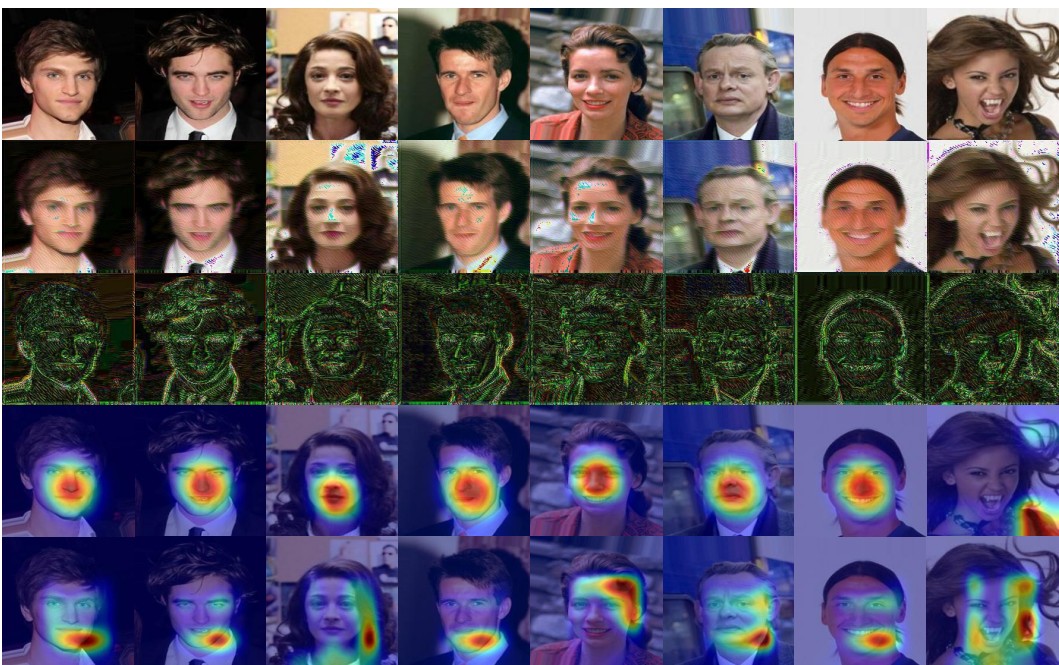

Figure 11: Visualization results on CelebA.

We also show some selected results to try to analyze why our method can improve the generalization ability of the model. Figure 13 is selected based on the predictions of BM and EM. Green boxes indicate that the samples are classified correctly, and red boxes indicate the wrong predictions. For action classification, what should really be concerned with is the action of the person itself rather than other things in the scene. Our method is able to better focus on the person, which results in better performance on the testing set where the environment changes. For example, for the vanilla model, mountain climbing is misclassified as fishing based on climbing ropes, and diving is misclassified as throwing based on spectator background, while EM can give the correct predictions.

In the proposed approach, AE and EM are alternately trained. As the training continues, AE may modify the samples in different ways, as shown in Figure 14 image below. The upper left corner is the original image, followed by the reconstruction results of AE at 10, 20, and 30 epochs.

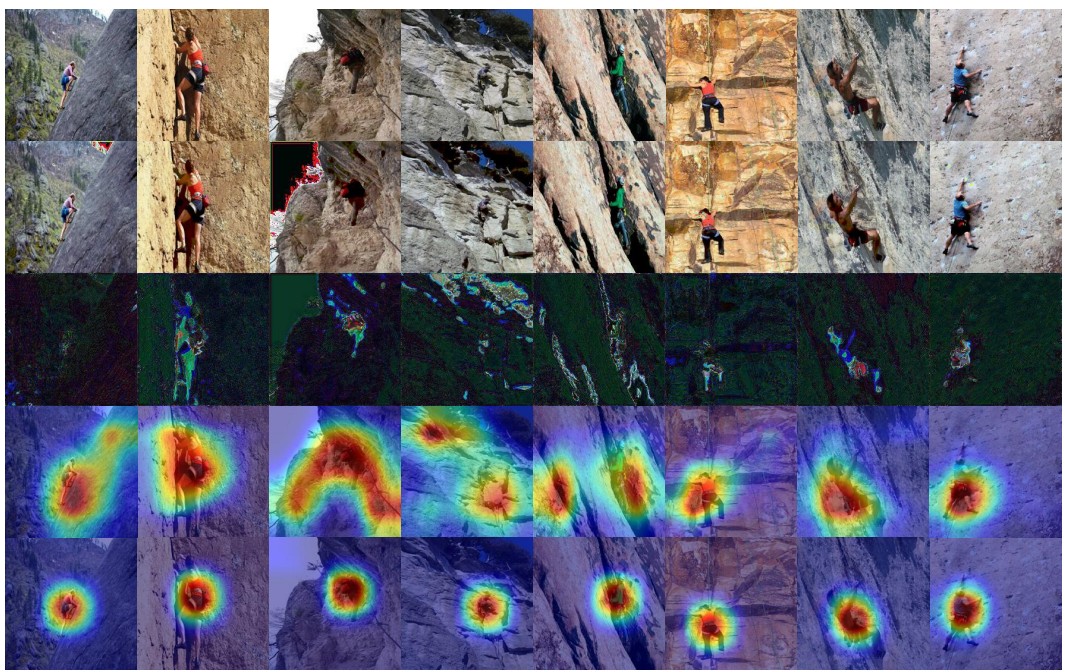

Figure 12: Visualization results on BAR.

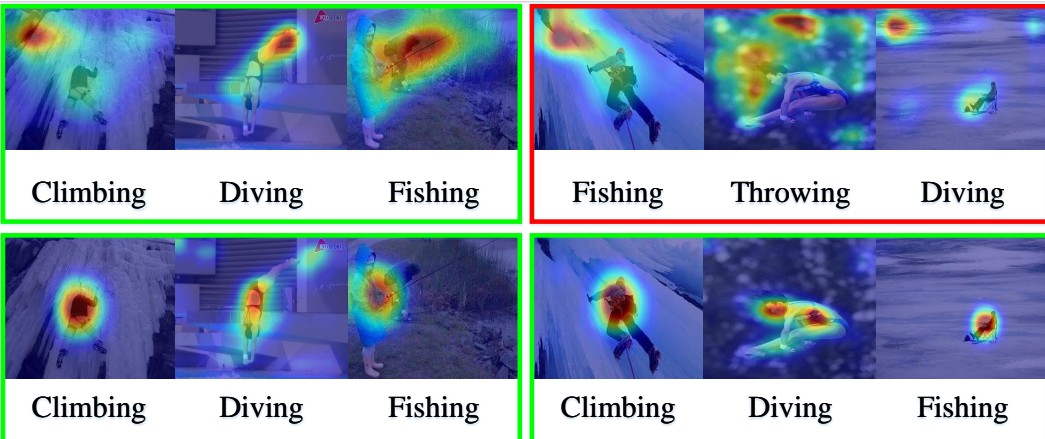

Figure 13: Some selected saliency maps. The upper row corresponds to vanilla and the lower row corresponds to EM. These saliency maps explain the difference when both are paired, and why EM has better generalization performance.

### A.7   More datasets results

We selected 10 categories of ImageNet to test the proposed approach in general scenarios. The 10 categories are n01608432, n01641577, n02106166, n02127052, n02389026, n02422106, n02690373, n03095699, n03100240, n03417042. Figure 15 presents the corresponding reconstruction results and saliency maps.

Table 6: Performance on ImageNet.

| Method | 0 | 1 | 2 | 3 | 4 | 5 | 6 | 7 | 8 | 9 | Avg. |
|---|---|---|---|---|---|---|---|---|---|---|---|
| Vanilla | 92.0 | 88.0 | 92.0 | 84.0 | 96.0 | 100.0 | 98.0 | 90.0 | 92.0 | 94.0 | 92.6 |
| Ours | 92.0 | 92.0 | 100.0 | 94.0 | 98.0 | 98.0 | 94.0 | 96.0 | 96.0 | 100.0 | 96.0 |

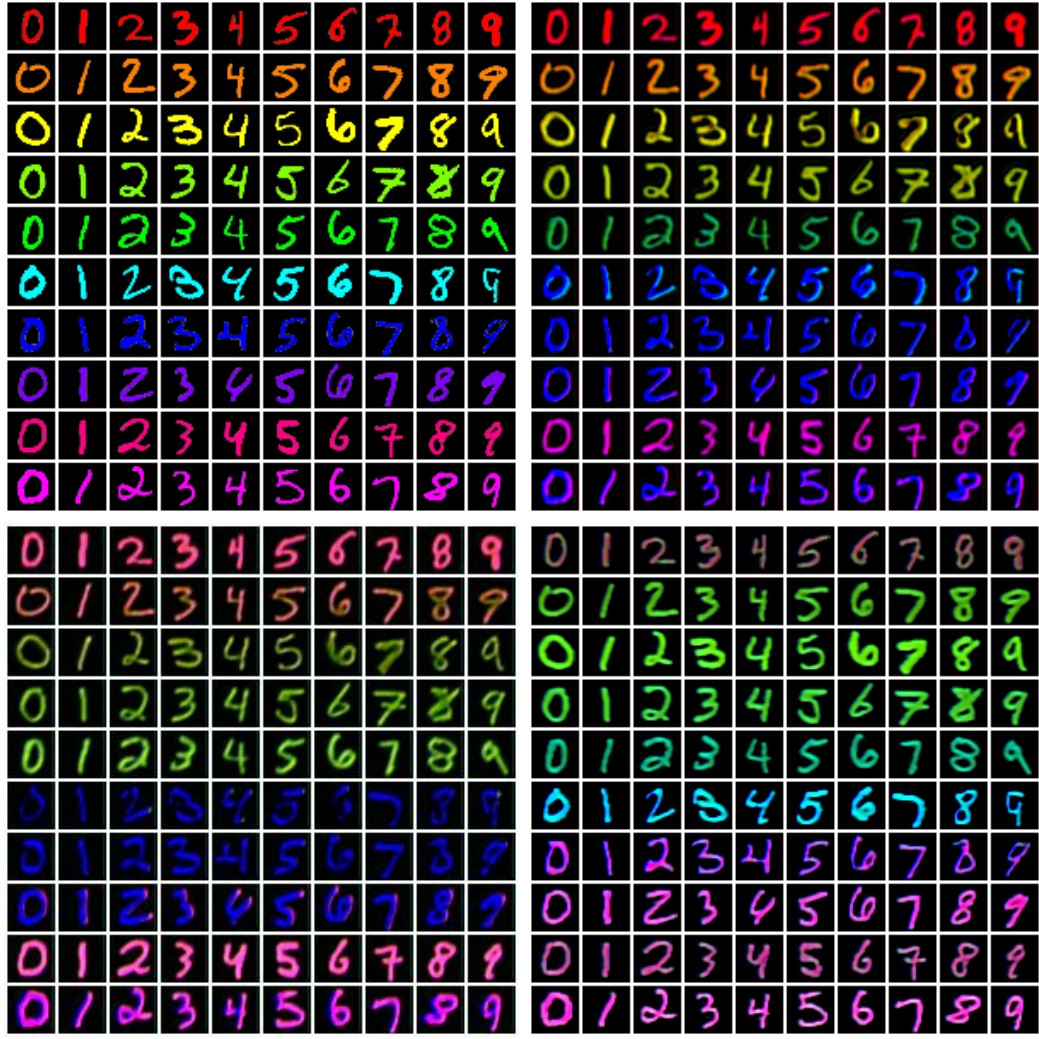

Figure 14: Reconstruction results on CMNIST.

In the ImageNet-9 dataset, we chose to modify the background to generate a challenging testing set (mixed-rand, mixed-next), and the results are shown in Table 7 and Figure 16.

Table 7: Performance on ImageNet-9.

| Method | Origin | Mixed-rand | Mixed-next |
|--------|--------|------------|------------|
| Vanilla | 90.08 | 66.32 | 62.59 |
| Ours | 90.50 | 70.21 | 67.73 |

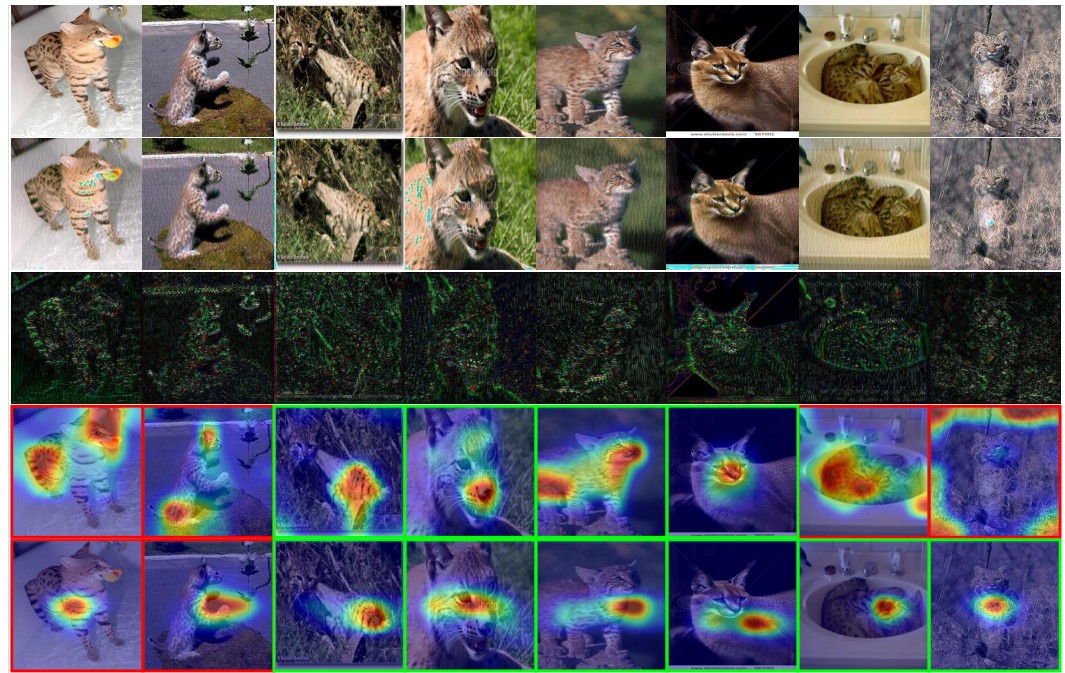

Figure 15: Visualization results on ImageNet.

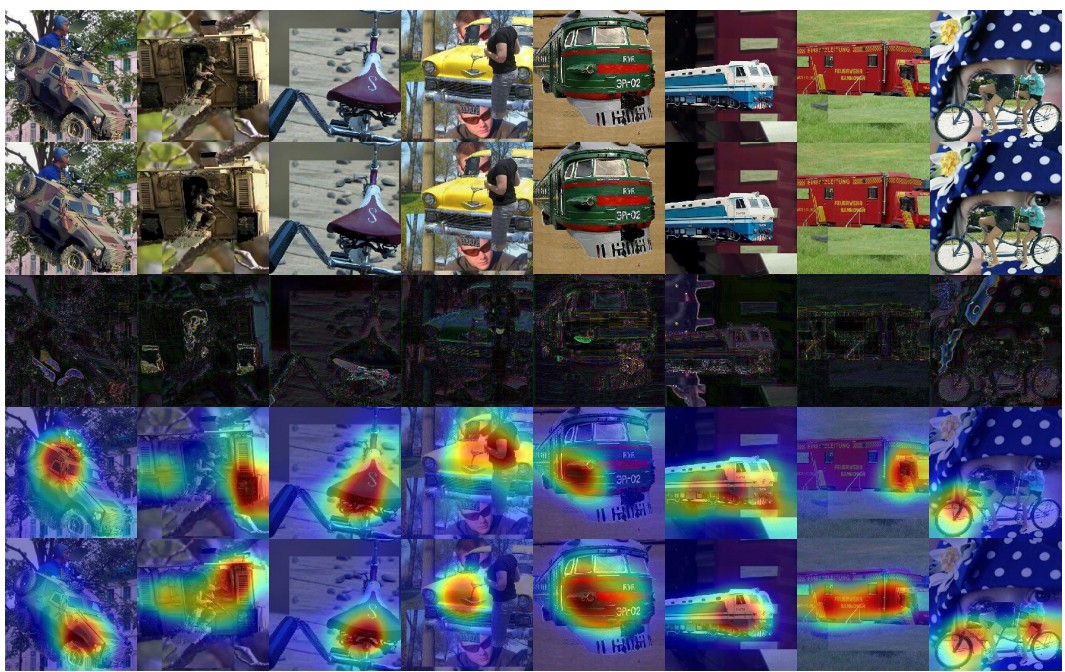

Figure 16: Visualization results on ImageNet-9.