# OpenReview forum: "Roadblocks for Temporarily Disabling Shortcuts and Learning New Knowledge"
_NeurIPS.cc/2022/Conference — NeurIPS 2022 Accept_

### Official Review · Reviewer_ius4 · 2022-07-11

**Rating:** 4
**Confidence:** 4
**Soundness:** 2 fair
**Presentation:** 1 poor
**Contribution:** 3 good

**Summary:**

The paper proposed a learning approach for robustness to shortcut learning: after training a base model (BM) on the biased dataset, the auxiliary autoencoder (AE) is trained to modify train images such that their reconstruction is close to the original input but the trained model  is outputting a uniform distribution over classes. Then another "exploration" model (EM) is trained on the modified inputs to predict the correct class. Thus, the intuition for the approach is that AE should remove the shortcuts from the inputs that BM learned, and EM will learn more diverse features. The approach was evaluated on ColorMNIST, CelebA and BAR benchmarks, and the authors provided additional analysis and visualizations of the features.

**Questions:**

I have listed by questions and concerns above in the "Strengths And Weaknesses". Summary:

1. Please provide the results for the ColorMNIST experiments with the MLP model (currently Roadblocks is using ResNet-18 while prior works are using MLP).

2. It would be helpful to see results on other benchmarks, especially a larger scale problem like ImageNet-9, as well as Dominoes (MNIST-CIFAR, MNIST-FashionMNIST, MNIST-MNIST), Waterbirds, CelebA [hair color, gender].

3. How are hyper-parameters chosen for the models (loss coefficients, weight decays, learning rates, hyper-parameters of alternate training, etc)? Is the validation split in-distribution or OOD (bias-conflicting)?

4. Please provide more details on the alternate training of the models.

5. It is not clear how the final predictions are made with the proposed model: is only EM model used?

6. Equation (3): why are you using MSE loss and not cross-entropy loss for classification?

**Limitations:**

Yes

**Strengths And Weaknesses:**

The paper is addressing an important problem of learning diverse features and training models robust to shortcuts, and the proposed approach shows promising results on CelebA and BAR, however, the clarity and presentation can be significantly improved, and the paper is missing implementation and method details. The results on CMNIST are not directly comparable to prior work due to a different model architecture. Also, experiments on large scale datasets would be needed to demonstrate the practicality of the proposed method.


### Originality
The paper is missing citations of the relevant literature on shortcut learning, simplicity bias, group robustness, and spurious correlations, e.g.:

Sagawa et al. Distributionally Robust Neural Networks for Group Shifts

Liu et al. Just Train Twice: Improving Group Robustness without Training Group Information

Shah et al. The Pitfalls of Simplicity Bias in Neural Networks

Xiao et al. Noise or Signal: The Role of Image Backgrounds in Object Recognition

Teney et al. Evading the Simplicity Bias: Training a Diverse Set of Models Discovers Solutions With Superior OOD Generalization

Nagarajan et al. Understanding the Failure Modes of Out-of-Distribution Generalization

Dagaev et al. A Too-Good-to-be-True Prior to Reduce Shortcut Reliance

Clark et al. Don't Take the Easy Way Out: Ensemble Based Methods for Avoiding Known Dataset Biases

Lee et al. Diversify and Disambiguate: Learning From Underspecified Data

and others. To the best of my knowledge, the proposed approach is novel. However, I encourage authors to explicitly discuss the connection of their work with Just Train Twice [Liu et al], Dagaev et al (see above) and LfF as all of these prior works are based on using 2 models where the second model is correcting the biases of the first one similarly to the proposed approach (while it additionally uses an AE to modify the inputs).


### Quality
1. ColorMNIST experiments: In the Appendix A.1 it is mentioned that ColorMNIST dataset is setup as in Lee et al [1]. It is also mentioned that ResNet-18 is used in all experiments. I noticed that the numbers for the Vanilla, EnD, ReBias, LFF, and LDD in Table 1 in this paper are taken from the Table 2 in Lee et al [1] (they match those numbers exactly). However, right below Table 2 in [1] it is mentioned that an MLP architecture is used in the experiments, while this paper uses ResNet-18 so the comparison in Table 1 is not valid.
(In general, I would also recommend that the authors explicitly say that which papers they took the numbers from for the prior methods in case they haven't re-run the methods.)

2. While the Roadblocks approach works well on CelebA [Heavy Makeup, Gender] and BAR, it would be very helpful to see the performance on a large scale problem, e.g. ImageNet-9 [2]. It would also be interesting to see the results on a broader range of benchmarks in shortcut learning / spurious correlations, e.g. Dominoes (MNIST-CIFAR, MNIST-FashionMNIST, MNIST-MNIST) [3, 4], Waterbirds, CelebA [hair color, gender] [5]. These benchmarks vary in the strength of the spurious correlation / bias in train data and the complexity of the shortcut feature.

3. How are hyper-parameters chosen for the models (lambdas loss coefficients, and also weight decays, learning rates, hyper-parameters of alternate training, etc)? Do authors use in-distribution validation set or bias-conflicting validation? This can have a high impact on the model's performance.

4. I would encourage the authors to provide the reconstructed images like in Figure 3 for other datasets besides CMNIST.


### Clarity
Clarity of the paper can be significantly improved. I strongly recommend the authors to add a brief datasets description to the main paper (e.g. the problem description, the nature of the bias / shortcut, the percentage of bias conflicting examples in train, etc). These details are not included in the paper, and some details become unclear without the context e.g. what "Ratio %" refers to in the Table 1.

It is mentioned that AE and EM are trained alternately, however, the details of this training are not described (i.e. are models alternately updated after each step or epoch?).

I strongly encourage the author to add the high-level details on the prior works to which Roadblocks method is compared to either in the Experiments section or in the Related work. In particular, it will be helpful to know how prior works compare to Roadblock method in terms of computational complexity, memory, etc.

Some paragraphs are unclear (e.g. lines 134-139, 172-176, 288-301) are not clearly written, lack details and context.

I also strongly encourage authors to add more details in Table and Figure captions in the next revision of the paper (e.g. Figure 6 doesn't have titles for subplots to indicate which dataset they correspond to).

It is not clear how the final predictions are made with the proposed model: is only EM model used? It is not clearly explained in the paper.

Line 47: "shortcuts are disabled temporarily, not forever", what does this mean? My understanding is that EM is trained on modified images outputted by AE so it is not clear what "temporarily, not forever" refers to.

Equation (3): why are you using MSE loss and not cross-entropy loss for classification?


### Significance
While the paper is addressing an important problem and achieves promising results on CelebA and BAR, more details and experiments are needed for publication.

**References**

[1] Lee et al. Learning debiased representation via disentangled feature augmentation.

[2] Xiao et al. Noise or Signal: The Role of Image Backgrounds in Object Recognition

[3] Shah et al. The Pitfalls of Simplicity Bias in Neural Networks

[4] Pagliardini et al. Agree to Disagree: Diversity through Disagreement for Better Transferability

[5] Sagawa et al. Distributionally Robust Neural Networks for Group Shifts

---

> ### Author Response · Authors · 2022-08-02
> **Response to Reviewer ius4**
>
> Thank you for acknowledging the importance of our research question and methodological novelty. Below we will respond to the shortcomings you raised.
>
>
> **Originality**
>
> Thank you for pointing this out. We think shortcut learning[1] is a good review, so we did not talk much about papers mentioned in shortcut learning. We have added a related discussion in the new version. We will also conduct further research.
>
> **Quality**
>
> **Q1** It was a severe carelessness. Thank you so much for pointing it out. For Table 1, we used the same three-layer MLP structure now, and we have updated Table 1. We also ran the methods listed in Table 1 and got close results. We did not get higher results and therefore used the results from the paper. We illustrated it in the table caption now.
>
> **Q2** We cannot add too many experiments due to time constraints. We conduct experiments on ImageNet, please refer to general response Q4; visualizations are in Appendix. We will experiment further on some interesting datasets. We believe that existing experiments have been able to demonstrate the effectiveness of our approach.
>
> **Q3** We tested different combinations of lambdas and selected the one that performed better, which is briefly described in the text. For other hyperparameters, we did not tune too much but adopted empirical settings. Both CMNIST and CelebA have validation sets with similar distributions to the training sets. BAR does not have a validation set, so we choose the results that perform better on the training set.
>
> **Q4** Alternate training means that models are alternately updated after each epoch. This process does not introduce hyperparameters that need to be tuned. Descriptions related to these issues have been added to the new manuscript.
>
>
> **Clarity**
>
> Thank you for your patient suggestion. We have made changes in the new version. Datasets description, alternately training, unclear paragraphs, details in Table, and Figure captions are added.
>
> **Q5** We discussed how the final predictions are made in the sum-up (lines 134-142). The experimental results shown in the Tables are the results of EM. The selection is based on the performance of BM and EM at training time (training set or validation set).
>
> Shortcuts are not necessarily wrong. In fact, the success of deep learning is largely due to the ability to find the most efficient features. So we prefer shortcuts to failures or pitfalls. If there's anything bad about shortcuts, that would be the reduction of the opportunity to learn more effective features. We want to provide a way to learn more effective features that are not necessarily better than shortcuts. Maybe the actual task can provide some priors to help us choose. Therefore, it is unreasonable and unnecessary to exclude shortcuts directly. We hope to clarify this with Figure 1. This is also the meaning of 'temporary'.
> Regarding computational complexity, our method also uses two classification models and may have similar results compared to some previous methods, so we did not compare them. Due to time constraints, there is no relevant content in this reply. We will consider discussing it later.
>
> **Q6** Please refer to general response Q3.
>
> **Significance**'
>
> Thank you for your positive assessment of the importance and novelty of our research. Also, thank you very much for your detailed and pertinent suggestions. We will try to describe them as best we can.
>
> [1] Geirhos, R., Jacobsen, JH., Michaelis, C. et al. Shortcut learning in deep neural networks. Nat Mach Intell 2, 665–673 (2020). https://doi.org/10.1038/s42256-020-00257-z

---

> > ### Author Response · Authors · 2022-08-08
> > **Looking Forward to Further Discussions**
> >
> > Dear reviewer ius4 :
> >
> > We sincerely thank you for the review and comments. You have given us a lot of professional and detailed advice before. And we have provided corresponding responses and results, which we believe have covered your concerns. We hope to further discuss with you whether or not your concerns have been addressed. Please let us know if you still have any unclear parts of our work.
> >
> > Best,
> > Authors of Paper 627

---

> > > ### Comment · Reviewer_ius4 · 2022-08-08
> > > **Response**
> > >
> > > Thank you for the answers to my questions and additional results!
> > >
> > > Q1. The authors addressed a technical flaw of the paper -- invalid comparison to prior works in Table 1, and updated the results, thus, I increased my score from 3 to 4. With the proper comparison, there is now a quite significant gap in performance compared to the prior method ReBias.
> > >
> > > Q2. I believe that current empirical evaluation is not sufficient for the paper to be published at NeurIPS, and I strongly encourage the authors to add experiments on a wider range of benchmarks for better understanding of the method: ImageNet-9 [2], Dominoes (MNIST-CIFAR, MNIST-FashionMNIST, MNIST-MNIST) [3, 4], Waterbirds, CelebA [hair color, gender] [5]. (please see my original review for references).
> > >
> > > Q3.
> > > > We tested different combinations of lambdas and selected the one that performed better...
> > >
> > > Hyper-parameter tuning strategy can influence the results quite a bit, in particular, whether one chooses the hyper-parameters based on average performance or performance on bias-conflicting split, so please clarify this in the next revision.
> > >
> > >
> > > I have read the other reviews and the responses, and while the paper is proposing an interesting approach, the empirical evaluation is limited and the clarity of the paper needs to be significantly improved, so I believe it is not ready for publication in the current form.

---

> > > > ### Author Response · Authors · 2022-08-09
> > > > **Glad to Receive Your Response.**
> > > >
> > > > Glad to receive your response.
> > > >
> > > > Q1. Thanks again for pointing out Q1 and increaseing the score.
> > > >
> > > > Q2. We would like to remind, as you also agree, that what we propose is a novel way to overcome shortcuts to learn more features. Our experiments show that it can handle different types of shortcuts.
> > > > We cannot validate all shortcuts, but experiments on CMNIST, ImageNet, CelebA, BAR, etc. datasets have covered a considerable number of cases.
> > > > Because the rebuttal time is limited, we cannot test the datasets you listed one by one, we further tested on ImageNet-9 and the results have been added in the appendix. We believe that among these several datasets, it is the most representative and challenging one.
> > > >
> > > > Q3. Thank you for your suggestion. We describe it in the appendix, and we will describe it in more detail if necessary.

---

### Official Review · Reviewer_T7g9 · 2022-07-11

**Rating:** 5
**Confidence:** 4
**Soundness:** 3 good
**Presentation:** 2 fair
**Contribution:** 3 good

**Summary:**

The authors propose an approach to address the problem of shortcut vulnerability of deep vision models in image classification tasks. For a trained, shortcut exploiting model (BM) the authors propose to train a model (AE) that learns to modify training samples in a minimal way while yielding a maximum entropy prediction for the BM model. Using these manipulated samples, an explorer model is trained (EM) in a supervised manner to predict the original ground-truth labels. The intuition behind the approach is that AE learns to manipulate input samples specifically to remove the shortcut information in the image in order to enforce the explorer model to learn new, actual class information and is hence less shortcut vulnerable.
Empirical evidence is provided for the CMNIST, CelebA and BAR datasets indicating the improved classification generalization. Further, the authors provide qualitative evaluation in form of an inspection of the manipulated images, as well as saliency evaluations comparing BM and EM which suggest that the AE model in fact removed the shortcut information and that the EM model attends to more meaningful image areas than BM.

**Questions:**

See weaknesses above.

**Limitations:**

See weaknesses above.

**Strengths And Weaknesses:**

In a nutshell, the work address a major, central problem of current ML The proposed solution to the problem is original and some interesting empirical results are provided. Still, I think some limitations / questions should be addressed more explicitly (see below). I would like to increase my rating in case stated limitations, questions (see below here under weaknesses) get addressed sufficiently.

Strengths
------------
The work addresses the problem of shortcut vulnerability of ML models in a supervised learning setup. This is of immense significans for the entire ML community.  As far as I can tell, the authors propose a novel, original approach to tackle this challenge in the effort to traina more shortcut robust model. The authors present their central idea in a clear, easy to access manner and present some quantitative evidence that supports the central claims of the paper (Tab 1, 2). Also of value are the provided qualiatitve evaluations indicating that the modification model (AE) works as intended on the investigated dataset CMNIST (Fig. 3) and a saliency comparison for BM and EM (Fig 5) on the BAR dataset, indicating that EM attends to more meaningful image areas than BM.

Weaknesses
----------------
Major 1

It is not clear to me what the AE actually learns and in which cases it actually learns to remove shortcut information. I think the provided intuition how AE should work is sound and the provided results support this intuition, however question about how AE actually works and how well it generalizes to other cases remain unclear and should be addressed:
* Why does AE not learn to produce (untargeted) adversarial noise? This would have the same effect as removing the shortcut, i.e., induce minimal change in the image and yield uniform classification. Please address in the text.
* Why does AE not learn to remove actual class information? This is especially relevant in case of low correlations (ratio) between the shortcut and the actual target class in the training dataset. In those cases, AE would probably learn to also remove the actual class information. Please discuss this point.
* More complicated shortcut structures might exist (shortcuts of shortcuts of shortcuts of ...). For this, see also [1] which should also be cited in intro / related work. This point seems partially addressed in L126-L133 but could be made more clear.
* Dependent on the design of the AE model, only rather specific types of shortcuts might be addressed with the used AE model. Also see [1] for more shortcut examples. For example, it is not clear that AE would be able to address the texture bias on ImageNet. Please also address why ImageNet was not selected as a dataset.
* To improve intuition about how AE works, please also provided AE modified samples for the BAR and CelebA dataset (similar to Fig 3 for CMNIST) and also provide saliencies for CMNIST and BAR (similar to Fig 5). Either Fig 3, 5 could be extended or additional images can be shown in the supplementary material.
* How cherry-picked are samples in Fig 3 and 5. Information in the text is missing.

[1] "DiagViB-6: A Diagnostic Benchmark Suite for Vision Models in the Presence of Shortcut and Generalization Opportunities", ICCV 2021

Major 2

Code should be made available to enable reproducibility. I could not find a code publishing statement in the paper.


Medium

* The role of the sum-up combination is not clear. Is this used somewhere in the experiments or is this merely a proposal for an open-set-recognition task? If of no practical relevance for this work it could be left out entirely.
* It should be noted somewhere that this training proceedure, similar to all data augmentation approaches, induces certain invariances in the EM model which might lead to unintended behavior on outliers.

---

> ### Author Response · Authors · 2022-08-02
> **Response to Reviewer T7g9**
>
> Thank you for acknowledging the importance of our research, protocols and experiments, which mean a lot to us. We will respond to your questions individually, hoping to clear some of your concerns.
>
> Below we respond to each of your points. Some of them are discussed in the updated manuscript now. If you think there is anything else that is more important to be put into the main text, please feel free to suggest it.
>
>
> **Major1-1 Why does AE not learn to produce (untargeted) adversarial noise?**
>
> Beacuse the goal and the corresponding loss function different. Please refer to general response Q3 for more discussion.
>
> **Major1-2 Why does AE not learn to remove actual class information?**
>
> What information the AE removes is determined by what information the BM has learned. If BM has learned the actual class information, it is also removed. At this point, our method selects BM as the final model.
> Unlike LFF, we do not share the idea that simple is bad. Therefore, we only temporarily disable shortcuts. Our approach provides a way to learn other potentially effective features. As for how to choose them, it should be determined by the situation. We suggest some sum-up strategies in lines 134-142. For the debiasing task, when the ratio is low enough, it is difficult to judge who is the target and who is the bias. In methods such as LFF, it is assumed that what is learned first is bias, and the method breaks down when the target attribute becomes an easily learned attribute. Therefore, we propose to decide based on the performance during training. Because in the training set, the prediction accuracy of the target feature must be higher than the bias accuracy.
>
> **Major1-3 More complicated shortcut structures might exist.**
> The proposed approach can continuously accumulate new EMs. However, in our experiments, the samples generated by the new AEs that keep both BM and EM at a loss often do not retain enough information. The debiasing task usually assumes only two, while in the conventional task, not only one feature is learned at a time. We do not observe the occurrence of complex shortcuts. We construct a three-attribute dataset to achieve this scenario, and with the second AE and EM, our approach can achieve the suppression of the first two attributes.
>
> **Major1-4 Dependent on the design of the AE model, only rather specific types of shortcuts might be addressed with the used AE model. Why ImageNet was not selected as a dataset.**
>
> We believe that the proposed framework can handle various shortcuts without priors, which is a significant contribution. If BM can extract features to complete classification, AE can disable the features to confuse BM. We are also trying other generative models to get better results.
> Please refer to general response Q4 for discussion about ImageNet.
>
> **Major1-5 To improve intuition about how AE works, please also provide AE modified samples for the BAR and CelebA and also provide saliencies for CMNIST and BAR.**
>
> We have made adjustments in the updated manuscript, thanks for your suggestion.
>
> **Major1-6 How cherry-picked are samples in Fig 3 and 5.**
>
> Figure 3 shows randomly selected samples for each category. Figure 5 is randomly selected from samples classified correctly by both models. Well-chosen saliency maps may mislead our perceptions, and I had similar concerns when I read the article myself. We put more saliency maps in Appendix, and our code will be open-sourced. Hope these reduce some of your worries.
>
> **Major2**
>
> We have indicated in the checklist and appendix that we will make the code public, so please do not worry about that.
>
> **Medium**
>
> Shortcuts are not necessarily a bad feature, which seems misleading. So we want to convey this point through sum-up.
>
> Our method does introduce some invariance, which is why it can disable shortcuts. We do not know much about data augmentation methods and look forward to having a more in-depth discussion with you.

---

> > ### Author Response · Authors · 2022-08-08
> > **Looking Forward to Further Discussions**
> >
> > Dear reviewer T7g9:
> >
> > We sincerely thank you for the review and comments. You have given many creative and constructive ideas about our work before. And we have provided corresponding responses and results, which we believe have covered your concerns. We hope to further discuss with you whether or not your concerns have been addressed. Please let us know if you still have any unclear parts of our work.
> >
> > Best, Authors of Paper 627

---

### Official Review · Reviewer_dvxL · 2022-07-11

**Rating:** 7
**Confidence:** 4
**Soundness:** 3 good
**Presentation:** 4 excellent
**Contribution:** 3 good

**Summary:**

A method for OOD generalization is proposed. To prevent a model (Explorer Model) from learning "shortcut features" (simple features which do not work in OOD setting~spurious correlations), an autoencoder is trained, which tries to fool a pretrained model (Blocked Model) as an additional loss. The explorer model is then trained on the reconstructed image, from which some of the shortcut features were hopefully removed by AE to fool the Blocked model.

**Questions:**

- Can we see some samples of CelebA reconstructed by the autoencoder?
- Can you discuss the relationship/tradeoffs with using a better autoencoder/pretrained autoencoder, which may limit the applicability of this method to harder datasets?

**Limitations:**

Relies on autoencoder reconstruction, which may be blurry and lower the prediction quality.

This method seems to implicitly perform some debiasing, which should hopefully have a positive impact in terms of making better predictions. However, this should be more extensively tested on a wide variety of tasks to see if the usual fairness/bias metrics are consistently improved.

**Strengths And Weaknesses:**

Strengths:
- Promising framework and approach, which does not require any external knowledge but relies on the interaction between autoencoder and blocked model.
- Interpretability: can visualize reconstructed images to gain insight (e.g. Figure 3 of CMNIST)
- Indirectly applicable to debiasing (by removing shortcut features)

Weaknesses:
- Relies on autoencoder reconstruction, which may be blurry and lower the prediction quality.

---

> ### Author Response · Authors · 2022-08-02
> **Response to Reviewer dvxL**
>
> Thank you for the compliment on our paper. It is a great encouragement for us.
>
> **Q1**
>
> In the updated manuscript, we present visualizations across datasets, including reconstructions and saliency maps. Please refer to Figure3, Figure5. Related discussions are in lines 181-191, 202-211. These Figures are randomly selected. More visualization results are in the appendix (Figures 11-15).
>
> **Q2 & L1**
>
> The requirements of our method for AE are actually not as high as those of other generative models. Because it only needs to make some changes based on the input without generating high-quality images from a simple input (noise or encoding). For regular datasets, the reconstruction results of the autoencoder are sufficient for classification with little impact. We also conducted experiments on ImageNet, and the results are as follows. Figures and some brief descriptions are included in Appendix A.7. Also, in Appendix A.3, you can see that we have constructed a scene with insufficient AE capability to conduct experiments, giving some interesting phenomena and analysis.
>
> **L2**
> Debiasing is not the core content, so we only discuss the basic debiasing tasks in this paper. We will further investigate the related work, and if necessary, we will discuss it in the appendix.

---

### Official Review · Reviewer_BFAY · 2022-07-12

**Rating:** 5
**Confidence:** 4
**Soundness:** 2 fair
**Presentation:** 1 poor
**Contribution:** 2 fair

**Summary:**

The paper tackles the problem of neural networks learning ‘simple’ features which cannot generalize well. It proposes to alleviate this problem of shortcut learning by generating synthetic data (“roadblocks”) in a manner which confuses the network while being close (in $l_2$ norm) to the original training data. A new model is trained on such examples, and it is hypothesized that this new model learns more robust features. The paper empirically shows that such models are more robust when there are spurious correlations in the data. The paper also presents visualizations of learnt features, as well as of the generated roadblocks, indicating that these do contribute to learning more robust features.

**Questions:**

1. How important is the distance metric for imposing loss between the generated images and the real images? Can a perceptual loss be used for the same?
2. How does the method compare against simple adversarial training?
3. Why was the MSE loss used for classification? Can cross-entropy give better results?

**Limitations:**

Some of the limitations are listed in the weaknesses above. A more thorough empirical evaluation on other OOD tasks would also be appreciated.


**Strengths And Weaknesses:**

Strengths -
1. The empirical results on the debiasing task are promising.
2. The method does not need any priors about the dataset in order to work.

Weaknesses -
1. The work misses several key references. Shortcut learning is very closely related to the problem of simplicity bias in neural networks [1], which is not cited in the work.
2. The method is also very similar to training with adversarial examples [2]. This line of work is neither cited nor compared against. On the mentioned datasets, the generated images indeed look very similar to $l_2$ or $l_\inf$ constrained adversarial attacks.
3. The paper proposes using the MSE loss for classification, which is not standard.
4. The writing of the paper can be improved significantly.


[1] - Shah, Harshay, et al. "The pitfalls of simplicity bias in neural networks." Advances in Neural Information Processing Systems 33 (2020): 9573-9585.
[2] - Ilyas, Andrew, et al. "Adversarial examples are not bugs, they are features." Advances in neural information processing systems 32 (2019).

---

> ### Author Response · Authors · 2022-08-02
> **Response to Reviewer BFAY**
>
> Thanks for your thoughtful review and constructive comments. I will respond to the weaknesses (**W**) and questions (**Q**) you mentioned.
>
> **W1**
>
> Thank you for pointing this out. We have discussed the literature you listed in the updated manuscript.
>
> **W2 & Q2**
>
> Our approach may be somewhat similar in form to adversarial training. However, there are clear differences in purpose and design. The intention of adversarial examples is to make predictions wrong, and untargeted attacks need to avoid correct predictions. However, the samples we construct through AE are intended to make predictions confusing, which means correct predictions and other possible categories are treated equally. We tested adversarial training on CMNIST, and the results are as follows. Related discussions are in lines 279-286 of the new version.
> | Ratio (%) | Vanilla | Adversarial(l2) |
> | --------- | ------- | --------------- |
> | 0.5       | 74.90   | 72.78           |
> | 1         | 65.77   | 64.14           |
> | 2         | 50.13   | 45.31           |
> | 5         | 33.07   | 30.47           |
>
> **W3 & Q3**
>
> MSE is closer to a distance metric than just classification loss. If only classification accuracy is considered, MSE may focus on unnecessary parts, but we believe that MSE is more in line with the need to suppress shortcuts. For example. the label is (1,0,0), there are 2 predictions, (0.8,0.1,0.1), (0.8,0.2,0). MSE think the former is better, while CE think they are the same. If we only aim at classification accuracy, the two should be the same. But we argue that the latter has higher confidence in an error class, which is more likely to lead to errors in tasks where shortcuts exist. This is not the first to use MSE loss with special considerations; for example, [1] discusses the application of MSE loss to imbalanced classification tasks.
>
> MSE is more prone to getting stuck in a local optimum, but in our task scenario, methods using CE have also gotten stuck in a local optimum. And our method keeps the model looking for new possibilities. Even if it falls into a local optimum, it is still likely to obtain a better solution under the overall framework of our method. We tried CE, and the results are not better.
>
> **Q1**
>
> Thanks for your very enlightening question.
> Perceptual Loss may also apply here. We did not do too complicated a design on the loss function. We think such a framework would make much sense for the community and we cannot wait to share this and be able to highlight it. In addition, we are concerned that some losses, such as Perceptual Loss, may have a more explicit feature preference orientation, which requires more thought and experimentation to verify.

---

> > ### Comment · Reviewer_BFAY · 2022-08-04
> > **Response to the Rebuttal**
> >
> > I thank the authors for a comprehensive response and revision of the manuscript. I still have a few lingering concerns, which I list below.
> >
> > 1. The response is missing the reference to the work which uses MSE loss for imbalanced classification. The example given by the authors is also not entirely clear to me. I am not sure why predictions having a lower entropy should be penalized more. Further, it is not even clear to me if models which exhibit shortcut learning actually have predictions following the pattern of logits in the example. Finally, it would be nice if the authors could share some concrete empirical results on the difference between the CE and MSE loss, since that would address the concerns.
> > 2. I appreciate the comparisons with adversarial training provided by the authors, but I would also be interested in knowing the details of the adversarial training approach considered by the authors, in particular the number of attack steps and the maximum norm of the adversarial perturbation. A well tuned adversary should be able to change the colour of the image on CMNIST while keeping the shape to be the same, so it is somewhat surprising to me that this does not lead to a better result.
> > I am inclined to change my score if the above concerns are addressed more concretely.

---

> > > ### Author Response · Authors · 2022-08-05
> > > **Glad to hear your further reply**
> > >
> > > **MSE**
> > >
> > > Here is the reference to the work which uses MSE loss for imbalanced classification.
> > >
> > > Kato S, Hotta K. MSE Loss with Outlying Label for Imbalanced Classification[J]. arXiv preprint arXiv:2107.02393, 2021.
> > >
> > > The example is used to illustrate that between predictions with the same cross-entropy loss, there are still differences that MSE can find.
> > > We believe that if a prediction has a higher response to a class other than the correct class, it means that it is more likely to be wrong on other test samples.
> > > Here are some experimental results on CMNIST using cross-entropy and MSE loss.
> > >
> > >
> > > | Ratio (%) | CE    | MSE   |
> > > | --------- | ----- | ----- |
> > > | 0.5       | 65.45 | 66.64 |
> > > | 1.0       | 80.98 | 82.04 |
> > > | 2.0       | 85.21 | 84.93 |
> > > | 5.0       | 88.72 | 88.65 |
> > >
> > > Using MSE is just a more intuitive option for us. The present results show no apparent difference in our method using CE and MSE. We will try to use CE as the loss on other datasets. If CE works better, we will make changes. Nevertheless, we would like to remind that this is not what we mainly want to share. We hope to provide a framework that suppresses shortcuts to learn more features.
> > > We believe that more suitable loss selection and training methods are waiting for us to discover under this framework. For example, reviewer BFAY suggested perceptual loss as a loss that measures the difference between the reconstruction and the original image.
> > > We think the proposed framework can work on more tasks. The design of loss will also vary. What we want to share the most is the idea of temporarily disabling shortcuts by modifying images and the design idea of roadblock loss.
> > >
> > >
> > > **Adversarial**
> > >
> > > We used *Fast Gradient Sign untargeted Adversarial Attack*, and the epsilon is 0.3 for l2 in the former experiment.
> > >
> > > We further experimented with different epsilons and also tried linf.
> > > Besides, we added the testing result on adversarial samples.  It can be seen that adversarial training has helped the model to obtain corresponding adversarial robustness.
> > >
> > > The table below was obtained on CMNIST (5pct).
> > >
> > > | epsilon  | 1.00E-03 | 5.00E-03 | 1.00E-02 | 5.00E-02 | 1.00E-01 | 5.00E-01 |
> > > | -------- | -------- | -------- | -------- | -------- | -------- | -------- |
> > > | l2-acc   | 68.45    | 66.13    | 71.03    | 73.37    | 71.59    | 76.69    |
> > > | l2-adv   | 68.44    | 66.06    | 70.97    | 73.61    | 70.97    | 73.19    |
> > > | linf-acc | 69.47    | 73.66    | 77.26    | 67.7     | 51.58    | 11.35    |
> > > | linf-adv | 69.44    | 73.14    | 74       | 52.08    | 35.85    | 11.35    |
> > >
> > > We provide some visualizations now (please refer to the new version of Appendix, Figure 8 & 9). We observed no appreciable behavior of modifying color or shape.
> > >
> > > We also conduct experiments on *Projeccted Gradient Descent*, but it seems difficult to obtain effective adversarial examples using MLPs. So we tried using ResNet-18 as the backbone.
> > > epsilon=0.03, k=4, alpha=0.03/4. The accuracy is 92.91 on CMNIST (5pct) and 88.54 on adversarial examples. (vanilla can achieve 93.34).

---

> > > > ### Author Response · Authors · 2022-08-08
> > > > **Looking Forward to Further Discussions**
> > > >
> > > > Dear reviewer BFAY:
> > > >
> > > > We sincerely thank you for the review and comments, which has helped us improve our paper. We are especially happy to receive your second round of responses, which gives us the opportunity to provide a better response. We have provided corresponding responses and results, which we believe have covered your concerns. We hope to further discuss with you whether or not your concerns have been addressed. Please let us know if you still have any unclear parts of our work.
> > > >
> > > > Best, Authors of Paper 627

---

> > > > > ### Comment · Reviewer_BFAY · 2022-08-09
> > > > > **Response to rebuttal**
> > > > >
> > > > > I thank the authors for their detailed response, and apologize for the delay. Some of my concerns have been satisfied, and I have updated the score appropriately.

---

### Author Response · Authors · 2022-08-02
**General response**

We thank the reviewers for their thoughtful and constructive review of our manuscript. We were encouraged to hear that the reviewers found the shortcut problem we discussed to be important (Reviewers T7g9, ius4) and that they view our methodology as novel (Reviewer ius4, T7g9) and effective (Reviewer BFAY, dvxL). In response to feedback, we provide a general response to points raised by multiple reviewers, and an new version of manuscript is uploaded. We will also respond to questions raised by each reviewer individually.


**Q1: What exactly did the AE learn?**

All reviewers are keen to note that the AE is at the core of our approach. Everyone asked some questions about AE. For example, the behaviour of AE on complex samples (Reviewers dvxL, T7g9, ius4)?
Intuitively, what AE learns is an elimination method for the knowledge used by BM. In the updated manuscript, we present visualizations across datasets, including reconstructions and saliency maps. Please refer to Figure3, Figure5. Related discussions are in lines 181-191, 202-211. These Figures are randomly selected. More visualization results are in the appendix (Figures 11-15).


**Q2: Why use MSE loss for classification?**

MSE is closer to a distance metric than just classification loss. If only classification accuracy is considered, MSE may focus on unnecessary parts, but we believe that MSE is more in line with the need to suppress shortcuts. For example. the label is (1,0,0), there are 2 predictions, (0.8,0.1,0.1), (0.8,0.2,0). MSE think the former is better, while CE think they are the same. If we only aim at classification accuracy, the two should be the same. But we argue that the latter has higher confidence in an error class, which is more likely to lead to errors in tasks where shortcuts exist. This is not the first to use MSE loss with special considerations; for example, [1] discussed the application of MSE loss to imbalanced classification tasks.
MSE is more prone to getting stuck in a local optimum, but in our task scenario, methods using CE have also gotten stuck in a local optimum. And our method keeps the model looking for new possibilities. Even if it falls into a local optimum, it is still likely to obtain a better solution under the overall framework of our method. We tried CE, and the results are not better.


**Q3: Difference with adversarial training?**

Our approach may be somewhat similar in form to adversarial training. However, there are clear differences in purpose and design. The intention of adversarial examples is to make predictions wrong, and untargeted attacks need to avoid correct predictions. However, the samples we construct through AE are intended to make predictions confusing, which means correct predictions and other possible categories are treated equally. We tested adversarial training on CMNIST, and the results are as follows. Related discussions are in lines 279-286 of the new version. We use *Fast Gradient Sign untargeted Adversarial Attack*, and the epsilon is 0.3.
| Ratio (%) | Vanilla | Adversarial(l2) |
| --------- | ------- | --------------- |
| 0.5       | 74.90   | 72.78           |
| 1         | 65.77   | 64.14           |
| 2         | 50.13   | 45.31           |
| 5         | 33.07   | 30.47           |

We also test different experimental setups and present some visualizations.
Please refer to Appendix (A.4) for more.

**Q4:Why ImageNet was not selected as a dataset?**

ImageNet does not have multi-label or recognized bias, which makes further analysis difficult. So we didn't try the ImageNet experiment before. We tested our method with 10 classes of ImageNet, and the results are as follows. Figures and some brief descriptions are included in Appendix A.7.
| Method  | 0    | 1    | 2     | 3    | 4    | 5     | 6    | 7    | 8    | 9     | Avg. |
| ------- | ---- | ---- | ----- | ---- | ---- | ----- | ---- | ---- | ---- | ----- | ---- |
| Vanilla | 92.0 | 88.0 | 92.0  | 84.0 | 96.0 | 100.0 | 98.0 | 90.0 | 92.0 | 94.0  | 92.6 |
| Ours    | 92.0 | 92.0 | 100.0 | 94.0 | 98.0 | 98.0  | 94.0 | 96.0 | 96.0 | 100.0 | 96.0 |


**Q5: Reproducibility and Writing?**

We note that the reproducibility of our method also raises some concerns.
We have indicated in the checklist and appendix that we will make the code public, so please do not worry about that.
We apologize if our writing caused some difficulties. We have clarified ambiguities raised in review comments in the updated manuscript and will further polish our writing.

[1] Kato S, Hotta K. MSE Loss with Outlying Label for Imbalanced Classification[J]. arXiv preprint arXiv:2107.02393, 2021.

---

> ### Author Response · Authors · 2022-08-09
> **Thank you all so much**
>
> We sincerely thank the reviewers and AC for their contributions.
>
> The reviewers asked many thoughtful questions, which inspired us a lot. Their meticulous review also helped us to better present our work.
>
> After discussion, most of the reviewer's concerns were resolved (2/4 reviewers raised their rating). We always welcome more questions, even after the discussion session has ended.
>
> This paper proposes a simple, novel and effective framework to help deep learning models learn more. It is significantly different from existing methods, such as adversarial training, negative correlation learning and other methods, and has certain advantages in some scenarios.
> We are looking forward to more excellent works based on this framework.

---

### Meta-Review · Area_Chair_2Us2 · 2022-08-26

**Recommendation:** Accept
**Confidence:** Less certain

**Metareview:**

The submission describes a new method to avoid the shortcut learning behaviour in DNNs. After the rebuttal and discussion, most of the reviewers are positive about this submission since the proposed method does not require prior knowledge about the dataset and the strong empirical results for the debasing task. On the negative results, the reviewers argue that the experimental evaluation is not thorough enough. Overall, AC recommends acceptance but asks the authors to perform more rigorous evaluation for the camera-ready version including the fair tuning of the hyperparameters.

**Award:**

No

---

### Decision · Program_Chairs · 2022-09-14

Accept